# Specific Multiomic Profiling in Aortic Stenosis in Bicuspid Aortic Valve Disease

**DOI:** 10.3390/biomedicines12020380

**Published:** 2024-02-06

**Authors:** Borja Antequera-González, Neus Martínez-Micaelo, Carlos Sureda-Barbosa, Laura Galian-Gay, M. Sol Siliato-Robles, Carmen Ligero, Artur Evangelista, Josep M. Alegret

**Affiliations:** 1Group of Cardiovascular Research, Pere Virgili Health Research Institute (IISPV), Universitat Rovira i Virgili, 43204 Reus, Spain; borja.antequer@urv.cat (B.A.-G.); mariacarmen.ligero@salutsantjoan.cat (C.L.); 2Cardiac Surgery Department, Hospital Vall d’Hebron (CIBERCV), Universitat Autonoma de Barcelona, 08035 Barcelona, Spain; 3Cardiology Department, Hospital Vall d’Hebron (CIBERCV), Universitat Autonoma de Barcelona, 08035 Barcelona, Spain; lauragaliangay@gmail.com (L.G.-G.);; 4Cardiology Department, Hospital Universitari Sant Joan de Reus, Universitat Rovira i Virgili, 43204 Reus, Spain

**Keywords:** aortic valve, aortic stenosis, mitochondrial dysfunction, oxidative stress, endothelial damage, metabolomics, transcriptome

## Abstract

Introduction and purpose: Bicuspid aortic valve (BAV) disease is associated with faster aortic valve degeneration and a high incidence of aortic stenosis (AS). In this study, we aimed to identify differences in the pathophysiology of AS between BAV and tricuspid aortic valve (TAV) patients in a multiomics study integrating metabolomics and transcriptomics as well as clinical data. Methods: Eighteen patients underwent aortic valve replacement due to severe aortic stenosis: 8 of them had a TAV, while 10 of them had a BAV. RNA sequencing (RNA-seq) and proton nuclear magnetic resonance spectroscopy (1H-NMR) were performed on these tissue samples to obtain the RNA profile and lipid and low-molecular-weight metabolites. These results combined with clinical data were posteriorly compared, and a multiomic profile specific to AS in BAV disease was obtained. Results: H-NMR results showed that BAV patients with AS had different metabolic profiles than TAV patients. RNA-seq also showed differential RNA expression between the groups. Functional analysis helped connect this RNA pattern to mitochondrial dysfunction. Integration of RNA-seq, 1H-NMR and clinical data helped create a multiomic profile that suggested that mitochondrial dysfunction and oxidative stress are key players in the pathophysiology of AS in BAV disease. Conclusions: The pathophysiology of AS in BAV disease differs from patients with a TAV and has a specific RNA and metabolic profile. This profile was associated with mitochondrial dysfunction and increased oxidative stress.

## 1. Introduction

Aortic stenosis (AS) is defined as the narrowing of the orifice of the aortic valve due to the stiffness and, frequently, calcification of the aortic valve leaflets, which causes a greater pressure gradient across the valve [1]. AS is one of the most common forms of heart disease, and its prevalence has increased to 10% of the population in the last eight decades [1]. Currently, there is no effective pharmacological treatment, and surgical processes such as transcatheter aortic valve replacement or aortic valve replacement are usually necessary [2]. Of the latter cases, at least 25% of patients 80 years of age or older who are referred for aortic valve replacement have a bicuspid aortic valve (BAV) [3]. BAV disease is the most common type of congenital heart disease, affecting up to 2% of the general population [4,5,6]. BAV disease is defined by a two-cusp morphology instead of the normal three-cusp morphology of the aortic valve, which causes an anomalous transvalvular flow [7,8,9]. This pathology is associated with quicker degeneration of the valve, with AS developing at a younger age than in individuals with a tricuspid aortic valve (TAV).

In individuals with a tricuspid aortic valve, known AS development risk factors include traditional cardiovascular factors such as smoking, metabolic syndrome, high low-density lipoprotein (LDL) cholesterol and hypertension, among others [10,11]. In BAV disease, the hemodynamic alterations and increased mechanical stress to which the anomalous aortic valve is already subjected are believed to also contribute to AS [10]. Despite the fact that there are few studies comparing AS pathophysiology in terms of aortic valve morphologies, some studies have suggested that they may not be equal [12,13]. Although not yet fully understood, endothelial dysfunction is thought to play a key role in AS due to mechanical stress, leaflet thickening, fibrosis and, in its later stages, calcification [1]. Alterations in lipoprotein metabolism and oxidative stress are also believed to affect the endothelium [1,2,3,4]. All of these factors seem to affect the hemostasis and permeability of the endothelium, thereby promoting activation of valvular interstitial cells (vICs) (myofibroblasts), which secrete proinflammatory cytokines, matrix metalloproteases and osteogenic factors [14,15,16,17]. This directly affects the function of the aortic valve, which loses elasticity and becomes more rigid [18]. The interplay between vECs and vICs, as well as the extracellular matrix (ECM), plays an important role in maintaining proper optimal valve function, and any alterations in any of its components can put it at risk [19].

Untargeted studies in the omics field have helped elucidate the pathophysiology of numerous diseases, and they have become powerful tools in the identification of new biomarkers [20,21]. Furthermore, untargeted data results can help to open pathways to identify the mechanisms that underlie these pathologies and to discover new therapeutic targets. In this study, aortic valves from patients with severe AS (TAV and BAV) who underwent aortic valve replacement were analyzed and 1H-NMR for metabolomics and RNA-seq for transcriptomics were performed on tissue samples from the aortic valves. The aim of this study was to further understand the differences between the two valves in the pathophysiology of AS and the mechanisms involved in the faster degeneration of the BAV.

## 2. Methods

### 2.1. Patients

We included 18 patients (8 TAV, 10 BAV) who underwent aortic valve replacement due to severe AS. Patient selection was adjusted for sex, and we tried to select younger patients with TAV due to the known younger age of AS patients with BAV. Patients for whom surgery was indicated due to ascending aorta aneurysm or ischemic heart disease were excluded. We also excluded patients with a grade of aortic regurgitation ≥III or chronic kidney disease (eGFR < 45 mL/min).

### 2.2. Aortic Tissue Extraction

In this study, 18 patients with an average age of 71 (57–83) years old were included. Aortic valve samples from patients who underwent aortic valve replacement were obtained, snap frozen in liquid nitrogen and stored at −80 °C at the biobank Vall d’Hebron Institut de Recerca. Samples were classified as TAV or BAV samples, both with severe AS.

### 2.3. Tissue Homogenization

Aortic valve tissues were transferred to a mortar under dry ice. Liquid nitrogen was quickly added to prevent denaturalization. Each sample was separately ground into powder and stored in 1.5 mL tubes. Homogenates were stored at −80 °C until used.

### 2.4. RNA-Seq

RNA was extracted using the PureLinkTM RNA Mini Kit (ref: 12183018A, Invitrogen, Waltham, MA, USA) and eluted with 30 μL of RNAse-free water. RNA was used for the creation of libraries. RNA was concentrated using a speedvac, and then libraries were created with a TruSeq small RNA Library kit (Illumina, San Diego, CA, USA) following the manufacturer’s instructions (Eurecat Centre Tecnològic de Catalunya-Centre for Omic Sciences (COS), Joint Unit University of Rovira i Virgili-EURECAT, 43204 Reus, Spain). DNA libraries were obtained and quantified by microfluidic electrophoresis on the Agilent TapeStation and the Agilent DNA High Sensitivity ScreenTape kit. Then, an equimolar pool was created at a concentration of 450 pM and later amplified by synthesis by adding labeled nucleotides with NextSeq2000 equipment from the Illumina platform. The generated libraries were read maintaining the direction of the transcripts and using 36 cycles. The resulting reads were separated per sample and converted to fastq format. Each sample was aligned to the human GRCh38 reference genome using HiSAT2, and the resulting alignments were used to assemble the transcripts with StringTie, which also generated matrix counts of transcripts and their respective genes. Features present in less than 25% of the samples or with a coefficient of variation <10% were removed. Differential expression analysis was performed using DESeq2.

### 2.5. Gene Enrichment Analysis

The complete list of significantly altered RNAs was introduced in the Enrichr-KG database (http://amp.pharm.mssm.edu/Enrichr/, accessed on 1 February 2023) [22,23], an online platform that provides summaries of numerous databases and visualizations of collective functions of gene lists [22]. Enrich-KG was utilized to explore pathways, functions and diseases related to these RNAs thanks to the combination of the WikiPathway, KEGG, Jensen Diseases, MGI and Gene Ontology (GO) databases. The representation of these results was performed using Prism v9 (GraphPad Software, La Jolla, CA, USA).

### 2.6. Lipid Profile and Low-Molecular-Weight Metabolite Quantification by 1H-NMR

For metabolomic analysis by nuclear magnetic resonance spectroscopy (1H-NMR), frozen homogenate aortic valve samples were sent on dry ice to Biosfer Teslab^®^, Reus, Spain. The 1H-NMR spectra were recorded on a BrukerAvance III 600 spectrometer (Billerica, MA, USA), as explained in [24].

A target set of 21 low-molecular-weight metabolites (LMWMs), including acetate, alanine, glucose, valine, isoleucine, leucine, gluconate, myo-inositol, lactate, glutamate, pyruvate, glutamine, 3-hydroxybutyrate, glycerol, glycine, formate, mannitol, choline, o-phosphocholine, creatine and histidine, were identified and quantified in the 1D Carr–Purcell–Meiboom–Gill (CPMG) spectra using a Dolphin adaptation [25,26].

Lipophilic extracts were also obtained using the BUME method with slight modifications [24,27]. Quantification of lipid signals in the 1H-NMR spectra was carried out with LipSpin [28], an in-house MATLAB-based software. Resonance assignments were made on the basis of values from the literature [29]. The 13 lipid species obtained by this NMR approach included cholesterol (free and esterified), triglycerides, unsaturated fatty acids (omega-6+omega-7 and omega-9), saturated fatty acids, polyunsaturated fatty acids, glycerophospholipids, phosphatidylcholine, sphingomyelin, lysophosphatidylcholine, linoleic acid and arachidonic + eicosapentaenoic acid.

### 2.7. Statistical Analysis and Multiomics Integrative Analysis

Data were analyzed using descriptive statistics. Quantitative variables were expressed as the median and 25–75 interquartile range and were compared using the nonparametric Wilcoxon rank-sum test. Qualitative or dichotomous variables were represented using percentages, and the chi-square test (χ^2^) and Fisher’s exact test were utilized to compare proportions and examine relationships.

In this study, we examined the predictive capacity of individual metabolites for aortic valve morphology using a separate model that was adjusted for age. Odds ratios were scaled to a 1-standard deviation (SD) increment in log-transformed metabolite concentration to facilitate comparisons across metabolites, and logistic regression was used to assess their associations with aortic valve morphology. The ggforestplot R package was employed for this analysis.

To identify key variables and patterns that distinguish between tricuspid and bicuspid valve morphologies, we performed an integrative analysis using the sPLS-DA in the mixOmics R package. Clinical data, low-molecular-weight metabolites, lipids and differentially expressed RNAs were integrated in this analysis. Relevant signatures that were highly correlated and associated with aortic valve morphology were visualized using Cytoscape version 3.9.1. These variables were subsequently included in a linear fitting model to predict the valve morphology. To evaluate the accuracy of the selected variables in discriminating between valve morphologies, we computed the area under the curve (AUC) and the 95% confidence interval of a receiver operating characteristics (ROC) curve. Patients were randomly assigned to training (50%) and test (50%) sets, and we performed a 10-fold cross validation with 100 replicates on the training data during the model construction process. The model was then tested on the hold-out data. All statistical analyses were conducted using R version 4.1.1.

### 2.8. Ethics

Written informed consent was obtained from all patients who participated in this study. This study was approved by the Research Ethics Committee (CEIm) of Hospital Universitari Vall d’Hebron.

## 3. Results

### 3.1. Clinical Characteristics of the Cohort

Clinical and echocardiographic characteristics are shown in Table 1. The results show that BAV patients were younger than TAV patients (Table 1). Furthermore, although it did not reach statistical significance, there was a trend toward a higher mean transaortic valvular gradient (mTAVG) and ascending aorta diameter in the BAV group than in the TAV group, as well as a higher prevalence of hypertension and diabetes mellitus in the TAV group than in the BAV group.

### 3.2. RNA Pattern Depends on Aortic Valve Morphology in AS

RNA-seq of 18 aortic valve samples from patients with AS and different aortic valve morphologies (BAV and TAV) was performed. After statistical analysis, the expression of 75 RNA molecules was found to be significantly different between the two groups (*p* value < 0.05). The complete data can be found in Appendix A. After performing multiple comparisons and applying an adjustment method, 5 of the 75 molecules remained statistically significant with an adjusted *p* value < 0.05. These five RNAs are described and represented in Figure 1.

For discovery purposes, the original 75 differentially expressed RNAs, before multiple comparisons, were used. Functional analysis of the differentially expressed RNAs was performed. The results showed that these RNAs were predominantly related to mitochondrial function and dynamics. Furthermore, the pathology databases also connected this RNA pattern to different cardiovascular diseases and processes (Figure 2).

### 3.3. The Metabolic Profile Seems to Be Specified by the Aortic Valve Morphology

The 1H-NMR analysis showed that some metabolites were differentially expressed in AS when compared to aortic valve morphology. Gluconate and choline levels were decreased in aortic valve samples from BAV individuals, while alanine was decreased in stenotic TAV patients (Table 2). These metabolites remained related to valve morphology after adjusting for age and adding myo-inositol and glutamate as metabolites decreased in BAV (Figure 3). However, no differences in lipid metabolites between the groups were found (Table 3).

### 3.4. Integrative Analysis of Clinical, 1H-NMR and RNA-Seq Data

The aim of the data integration was to find a pattern of variables that were highly associated with each other and could classify patients according to their valvular morphology. The data used for integration were as follows: clinical data, low-molecular-weight metabolites, lipids and differentially expressed RNAs determined by RNA-seq. Through integration, the selected variables formed the network shown in Figure 4A. The PLS-DA represented in Figure 4B shows the potential of the selected variables to distinguish between patients based on their aortic valve morphology.

The previously selected variables were included in a multivariate analysis to validate the proposed integrational model. The model was developed using 50% of the patients as training data, and the performance of the model was assessed using the remaining 50% of samples as a test set. Figure 5 illustrates the contribution of each variable in the classification of patients based on their valve morphology. The ROC curve depicted an area under the curve (AUC) of 0.9 (95% CI = 0.67–1), indicating high predictive accuracy. The model demonstrated an overall accuracy of 0.89, a sensitivity of 1, a specificity of 0.75, a *p* value of 0.041 and an out-of-bag (u) error value of 0.28, as shown in Figure 5.

## 4. Discussion

In the present study, we found a specific metabolic profile for AS patients with BAV as compared to TAV patients by 1H-NMR in aortic valve tissues. Metabolic data showed that gluconate, myo-inositol, glutamate and choline were downregulated in BAV individuals compared to TAV individuals. On the other hand, alanine was upregulated. The results of RNA-seq in aortic valve tissue also showed a different RNA pattern when comparing both groups, which was functionally related to mitochondrial dysfunction and dynamics. Furthermore, integration of the clinical 1H-NMR and RNA-seq data demonstrated that the pathophysiology of AS may be specified by the aortic valve morphology and that the mechanisms that underlie this pathology are not the same between BAV and TAV patients.

Abnormal aortic valve morphology is known to cause hemodynamic alterations. Although few studies have focused on vECs in BAV disease, some have noted the activation of these vECs by abnormal shear stress patterns. Paracrine signaling, calcification and catabolic enzyme secretion have also been associated with activity of vECs. Similar to what happens in the aortic media, these changes in hemodynamics could be capable of altering correct vEC functioning [30,31,32]. Altered blood flow can have different effects on endothelial cells, including inflammation, increased oxidative stress and mitochondrial function, due to mechanotransduction [33,34]. Furthermore, abnormal valve dynamics, such as those in BAV disease, could also influence this process by the increased mechanical stress the leaflets are submitted to during opening [35]. Endothelial cells can sense and transduce the increased shear stress into different intracellular biochemical signals [36]. This could play a key role in AS progression when combined with the highly discussed predisposition toward endothelial dysfunction present in BAV disease [37]. Furthermore, some studies have found that shear stress can play a role even in mitochondrial homeostasis in endothelial cells by controlling ATP generation and mitochondrial membrane potential [38,39]. Although mitochondrial dysfunction has been discovered in numerous cardiovascular diseases, no definitive answers have been found in BAV disease [39].

Mitochondrial dysfunction is present in many diseases, and, recently, numerous studies have identified its key role in many cardiovascular diseases and have proposed it as a potential therapeutic target. Mitochondria are crucial organelles that not only function in ATP production but are also the primary source of reactive oxygen species (ROS), triggering oxidative stress, and they can regulate cell death and survival [40,41]. Mitochondria are not static organelles; they are constantly subjected to an equilibrium of fusion and fission termed mitochondrial dynamics [42]. Both processes have an equilibrated interplay in physiological conditions that support cell and mitochondrial homeostasis. Nevertheless, in pathological conditions, these processes have been observed to be imbalanced [43]. Mitochondrial damage has also been related to alterations in metabolism [44]. In our study, gluconate, myo-inositol, glutamate and choline were observed to be downregulated in BAV individuals with AS compared to TAV individuals with AS. The expression alterations of these molecules have previously been related, to a greater or a lesser extent, to mitochondrial damage or dysfunction [45,46]. Specifically, myo-inositol has recently been found to serve as an inhibitor of mitochondrial fission [46]. On the other hand, alanine was found to be upregulated in BAV patients compared to TAV patients. High levels of alanine have been related to a reduction in ATP generation in the mitochondria, enhancing oxidative stress [47]. Therefore, it seems that the metabolic profile present in BAV disease is related to altered mitochondrial dynamics and increased oxidative stress. However, when comparing both groups, we did not find any differences in the lipid profile. Lipid metabolism has been discussed as a key player in aortic stenosis in general, but not many omic studies comparing both aortic valve morphologies have been published [10,16,20,48]. Although we cannot discredit lipid metabolism as a key player in aortic stenosis, it did not seem to differ that much between both aortic valve morphologies.

RNA-seq also showed a differential RNA profile between the BAV and TAV groups with severe AS. The five most significant RNAs were presented, and some of them had already been linked with alterations in mitochondrial homeostasis. COX7C, the most significantly altered RNA, corresponds to a subunit of cytochrome c oxidase, the terminal component of the mitochondrial respiratory chain, which drives oxidative phosphorylation [49]. This nuclear-encoded subunit is hypothesized to function in the regulation and assembly of the complex [50]. The second most differentially expressed RNA, MTND5P10, is a mitochondrial pseudogene whose function remains unknown. FCGR2a plays a key role in immune processes and is known to mediate changes in the plasma membrane potential of mitochondria and inflammation [51]. The other two most significant RNAs, SCARNA2 and RNU12, were less abundant but also bibliographically related to mitochondria: SCARNA2 is believed to play a role in the miR-342-3p-EGFR/BCL2 pathway [52], which controls mitochondrial apoptosis, and RNU12 also seems to be associated with altered mitochondrial electron transport in other pathologies [53].

Omics has already been used in the study of the pathophysiology of AS and its progression to calcific aortic valve disease (CAVD), mainly in TAV patients [20]. Some studies have observed a different proteomic architecture in the ECM of valve cells in CAVD compared to healthy control valves [50]. Furthermore, the metabolomic profile was also found to be AS progression dependent, where lysophosphatidic acid was positively associated with the AS grade [48]. A review with a multiomic focus also highlighted the different patterns found in omic sciences for AS [20]. Nevertheless, few comparative assays have compared AS development according to aortic valve morphology. A transcriptomic analysis showed similar mechanisms in TAV and BAV, including increased inflammation, downregulation of NOTCH-1 signaling and ECM alterations [12]. Nevertheless, dysregulation of fetal gene programs was present, demonstrating that, although similar, AS development in BAV disease differs from that in TAV [12]. In our study, a larger sample size for transcriptomics combined with metabolomics helped us discover a multiomic-specific profile for AS in BAV disease. This helped us connect AS in BAV disease to mitochondrial dysfunction.

Abnormal bicuspid aortic valve dynamics, including opening restrictions and blood flow alterations, promote high shear stress and turbulent eccentric flow, which could stimulate endothelial dysfunction in the aortic valve leaflets [7,8,9,37]. We hypothesize that these dynamic alterations combined with the predisposition of endothelial dysfunction inherent to BAV disease could alter vECs, including correct mitochondrial function, partly through mechanotransduction pathways. These alterations could be part of the explanation for the increased oxidative stress and endothelial dysfunction observed in BAV disease [34,37].

We should recognize some limitations. The sample size for aortic valve tissues was scarce. However, it was the sample estimated for obtaining rigorous transcriptomics and metabolomics. Age differences between both groups were also present, something almost inevitable in a comparison between patients with AS on TAV vs. BAV. Therefore, the metabolic profile was adjusted by age, and a complete integration of clinical, metabolic and transcriptomic data was performed. Furthermore, no subsequent studies analyzing mitochondrial dysfunction or dynamics in vivo or in vitro were performed, as our initial study was only for discovery, not validation. Mitochondrial dysfunction and oxidative stress could also be present in TAV. In this study, we identified differences related to valve morphology. Mitochondrial dysfunction should be studied specifically in BAV disease with AS in larger cohorts, and its utility as a new therapeutic target should also be assessed.

## 5. Conclusions

Aortic valve tissue samples from AS patients with BAV disease have different metabolic and RNA profiles than those from TAV individuals. This suggests that the pathophysiology of AS differs between BAV individuals and TAV individuals. In BAV disease, AS was related to increased oxidative stress and mitochondrial dysfunction, which could be directly related to the endothelial dysfunction and hemodynamic alterations present in this disease. New therapeutic targets focusing on rescuing mitochondrial and endothelial function should be assessed to evaluate their utility in slowing aortic valve degeneration in BAV individuals.

## Figures and Tables

**Figure 1 biomedicines-12-00380-f001:**
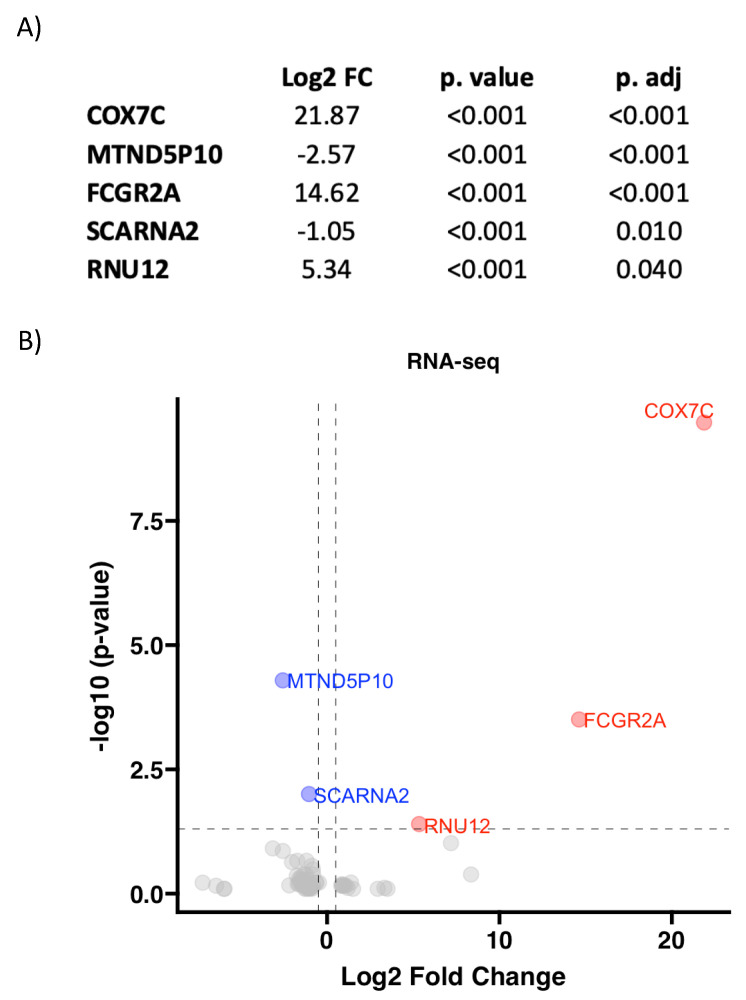
Five most significant RNAs in the TAV vs. BAV RNA-seq comparison. Data are presented in (**A**) and can be easily visualized in the volcano plot (**B**), where the overexpressed RNAs are represented as red dots and the downregulated RNAs are represented as blue dots.

**Figure 2 biomedicines-12-00380-f002:**
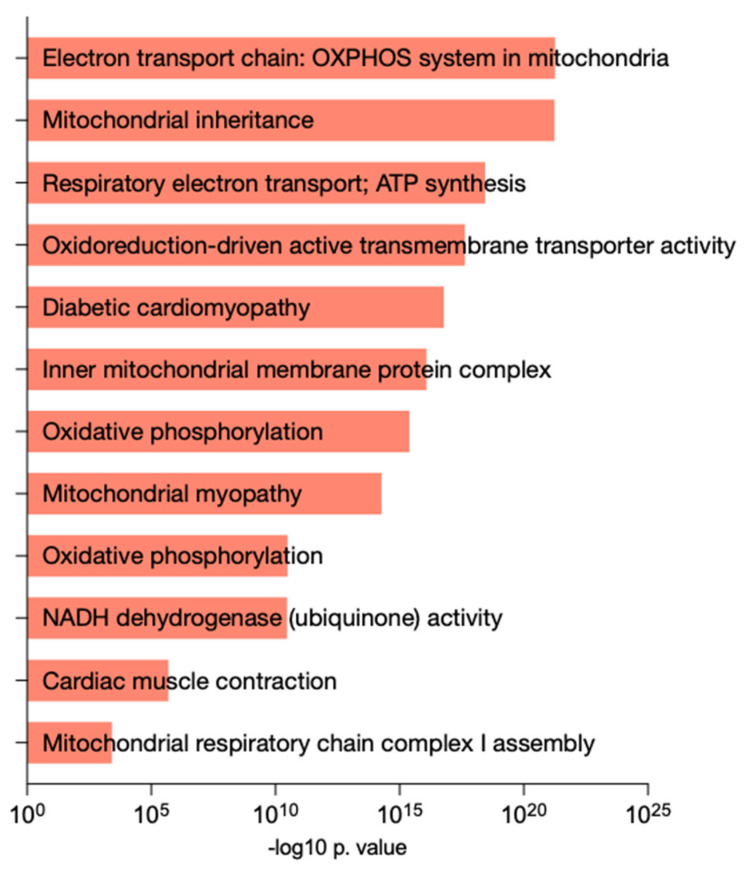
Visualization of RNA enrichment analysis with different databases. OXPHOS: oxidative phosphorylation; ATP: adenosine triphosphate; NADH: nicotinamide adenine dinucleotide (reduced).

**Figure 3 biomedicines-12-00380-f003:**
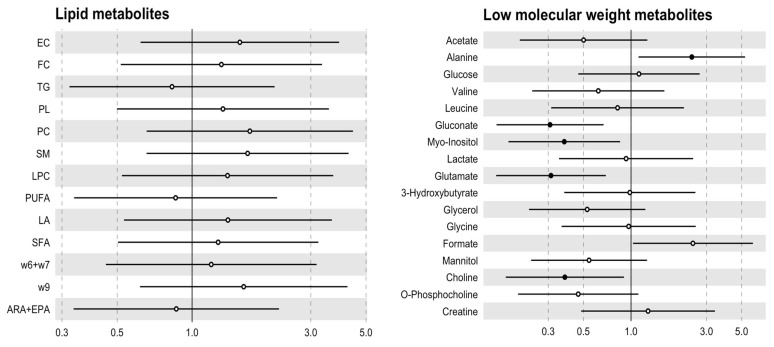
Association between lipid profile and low-molecular-weight metabolites with age-adjusted valve morphology. Odds ratio for aortic valve morphology (95% CI) per 1-SD increment in metabolite concentration. The black circle indicates a *p* value < 0.05. EC: esterified cholesterol; FC: free cholesterol; TG: triglycerides; PL: phospholipids; PC: phosphocholine; SM: sphingomyelins; LPC: lysophosphatidylcholine; PUFA: polyunsaturated fatty acids; LA: linoleic acid; SFA: saturated fatty acids; w6 + w7: omega 6 and 7; w9: omega 9; ARA + EPA: arachnoid acid + eicosapentanoic acid.

**Figure 4 biomedicines-12-00380-f004:**
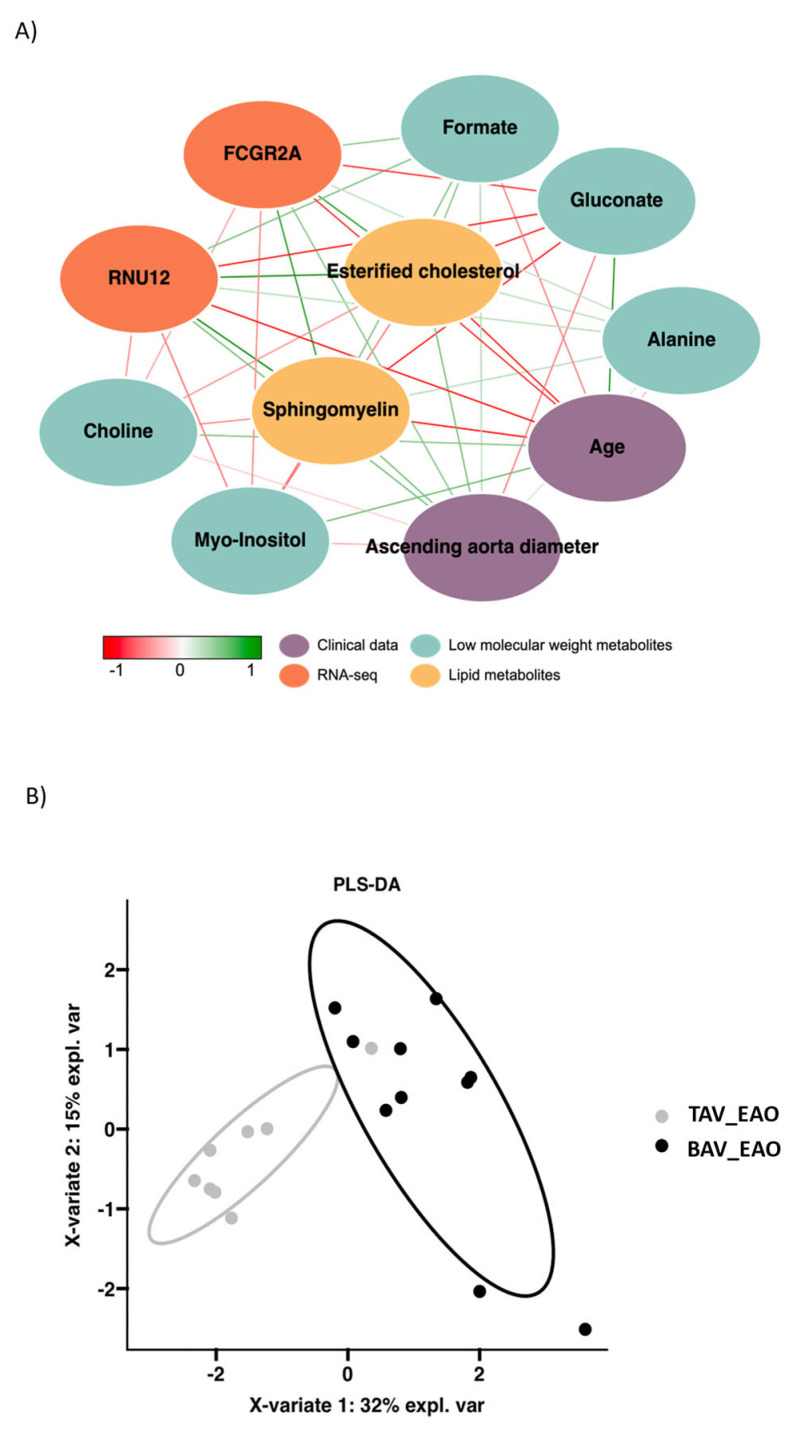
Visualization of data integration. (**A**) Network with parameters symbolized with nodes classified by color and associations with each other, represented with red or green lines. (**B**) PLS-DA with X variables explaining 32% and 15% of the cases divided by aortic valve morphology.

**Figure 5 biomedicines-12-00380-f005:**
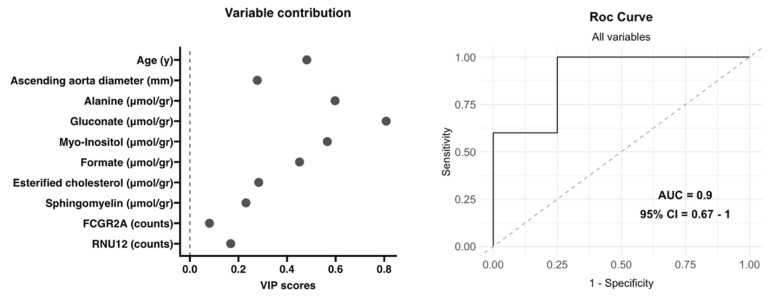
Evaluation of the multiomics integration model. This model included the variables shown on the left with their contributions (VIP scores) and a ROC curve that classified aortic valve morphology on the right.

**Table 1 biomedicines-12-00380-t001:** Clinical characteristics of the cohort.

	TAV (*n* = 8)	BAV (*n* = 10)	*p* Value
Sex			1.000
Female	2 (25.0%)	3 (30.0%)	
Age (year)	77.5 [76.8;78.5]	65.5 [61.2;73.2]	0.008
Weight (kg)	78.0 [66.2;106]	75.5 [66.5;82.8]	0.699
Height (cm)	168 [162;172]	168 [156;176]	0.948
BMI (kg/m^2^)	27.5 [26.4;36.1]	26.9 [25.5;28.1]	0.439
Hypertension	7 (87.5%)	4 (40.0%)	0.066
Diabetes mellitus	6 (75.0%)	2 (20.0%)	0.054
Dyslipidemia	6 (75.0%)	3 (30.0%)	0.153
Smoking			0.588
Ex	2 (25.0%)	1 (10.0%)	
No	6 (75.0%)	7 (70.0%)	
Yes	0 (0.00%)	2 (20.0%)	
Statins treatment	5 (62.5%)	3 (30.0%)	0.342
Renal insufficiency	1 (12.5%)	1 (10.0%)	1.000
Ischemic heart disease	5 (62.5%)	2 (20.0%)	0.145
mTAVG (mmHg)	45.0 [42.5;50.0]	55.5 [49.0;62.8]	0.062
AR (≤II)	1 (12.5%)	4 (40.0%)	0.348
LVDD (mm)	48.0 [45.0;52.0]	46.0 [40.8;48.0]	0.447
LVSD (mm)	33.0 [29.5;37.5]	30.0 [24.0;35.0]	0.306
LVEF (%)	60.0 [46.8;64.6]	60.5 [56.2;69.5]	0.475
Ascending aorta (mm)	36.5 [33.5;38.8]	41.5 [37.8;43.0]	0.068
Aortic root (mm)	36.0 [31.0;37.0]	36.5 [34.0;42.8]	0.349

BMI: body mass index; mTAVG: mean transaortic valvular gradient; AR: aortic regurgitation; LVDD: left ventricular diastolic diameter; LVSD: left ventricular systolic diameter; LVEF: left ventricular ejection fraction.

**Table 2 biomedicines-12-00380-t002:** Low-molecular-weight metabolite analysis by 1H-NMR.

Metabolites	TAV (*n* = 8)	BAV (*n* = 10)	*p* Value
Acetate (μmol/gr)	0.79 [0.18;1.43]	0.50 [0.21;0.67]	0.183
Alanine (μmol/gr)	0.08 [0.06;0.13]	0.15 [0.11;0.19]	0.046
Glucose (μmol/gr)	1.05 [0.83;1.83]	1.48 [0.74;1.73]	0.753
Valine (μmol/gr)	0.07 [0.07;0.08]	0.07 [0.06;0.08]	0.549
Isoleucine (μmol/gr)	0.03 [0.03;0.04]	0.04 [0.03;0.06]	0.201
Leucine (μmol/gr)	0.06 [0.04;0.07]	0.05 [0.04;0.07]	0.655
Gluconate (μmol/gr)	0.65 [0.45;1.19]	0.27 [0.09;0.44]	0.045
Myo-Inositol (μmol/gr)	0.38 [0.36;0.42]	0.23 [0.19;0.31]	0.188
Lactate (μmol/gr)	2.93 [2.08;3.76]	2.96 [2.29;4.17]	0.929
Glutamate (μmol/gr)	0.38 [0.30;0.45]	0.25 [0.13;0.31]	0.126
Pyruvate (μmol/gr)	0.02 [0.01;0.02]	0.02 [0.01;0.02]	0.462
Glutamine (μmol/gr)	0.28 [0.19;0.29]	0.23 [0.18;0.31]	0.655
3-Hydroxybutyrate (μmol/gr)	0.04 [0.03;0.18]	0.08 [0.07;0.10]	0.380
Glycerol (μmol/gr)	0.04 [0.03;0.07]	0.04 [0.02;0.05]	0.286
Glycine (μmol/gr)	0.11 [0.06;0.14]	0.14 [0.09;0.17]	0.657
Formate (μmol/gr)	0.16 [0.14;0.19]	0.22 [0.18;0.27]	0.050
Mannitol (μmol/gr)	0.92 [0.56;1.43]	0.34 [0.26;0.75]	0.167
Choline (μmol/gr)	0.08 [0.06;0.09]	0.05 [0.04;0.06]	**0.041**
O-Phosphocholine (μmol/gr)	0.16 [0.09;0.16]	0.10 [0.08;0.12]	0.149
Creatine (μmol/gr)	0.04 [0.04;0.05]	0.04 [0.03;0.07]	0.886
Histidine (μmol/gr)	0.08 [0.08;0.08]	6.47 [4.02;8.91]	0.221

**Table 3 biomedicines-12-00380-t003:** Lipid profile quantification by 1H-NMR.

Lipids	TAV (*n* = 8)	BAV (*n* = 10)	*p* Value
Esterified cholesterol (μmol/gr)	61.3 [53.5;73.5]	76.1 [68.6;93.0]	0.214
Free cholesterol (μmol/gr)	43.5 [36.9;47.8]	51.0 [35.4;56.7]	0.477
Triglycerides (μmol/gr)	12.0 [9.29;14.2]	15.3 [9.77;19.4]	0.594
Glycerophospholipids (μmol/gr)	10.8 [10.3;11.1]	11.5 [10.8;12.0]	0.286
Phosphatidylcholine (μmol/gr)	8.23 [7.48;9.45]	9.57 [7.83;10.3]	0.248
Sphingomyelin (μmol/gr)	13.7 [11.6;15.1]	17.7 [12.2;22.2]	0.214
Lysophosphatidylcholine (μmol/gr)	1.76 [1.60;1.92]	2.12 [1.49;2.20]	0.328
Polyunsaturated fatty acids (μmol/gr)	129 [115;158]	128 [110;143]	0.790
Linoleic acid (μmol/gr)	53.6 [47.9;56.9]	59.1 [48.9;68.7]	0.286
Saturated fatty acids (μmol/gr)	77.2 [71.6;81.2]	81.1 [67.8;91.8]	0.424
Omega-6 and fatty acids (μmol/gr)	16.9 [13.6;18.2]	19.6 [14.8;21.6]	0.374
Omega-9 fatty acids (μmol/gr)	49.4 [46.3;72.8]	69.9 [57.7;84.5]	0.248
ARA + EPA (μmol/gr)	16.1 [15.2;22.1]	16.3 [15.4;19.6]	0.790

ARA: arachidonic acid; EPA: eicosapentaenoic acid.

## Data Availability

The data presented in this study are available on request from the corresponding author. The data are not publicly available due to ethical reasons.

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
