# Peer review of "Specific Multiomic Profiling in Aortic Stenosis in Bicuspid Aortic Valve Disease"

_biomedicines, 2024, doi:10.3390/biomedicines12020380_

Round 1

Reviewer 1 Report

Comments and Suggestions for Authors

This is a very detailed paper.

However, it does not address mechanism

Thus, there is no attempt at explanation in the differences in Table 2.

Also, the lack in differences in lipid metabolism (Fig.3) should be discussed.

In the Discussion, no mentioned of the possible differences in shear stress derangement between  

TAV and BAV is given.

Also mitochondrial dysfunction and oxidative stress are also present in TAV as well as BAV.

Author Response

This is a very detailed paper.

However, it does not address mechanisms.

Thank you very much for your honest review. The goal of this study was to discover, thanks to the combination of different omics, new approaches in aortic stenosis in bicuspid aortic valve (BAV) disease. This study did not aim and was not designed to analyze the concrete pathways/mechanism that produce these effects. Nevertheless, we think of this manuscript as an starting chapter of research for new therapeutic targets in BAV disease but also to help better understand its development. Further studies using specific methodology will analyze the mechanisms involved.

Thus, there is no attempt at explanation in the differences in Table 2.

Thank you for your comment. We address the different expression of some metabolites between our groups through the whole results and discussion section. Furthermore, we were able to link these expression changes bibliographically to mitochondrial damage in the discussion section (lines 316-325).

Also, the lack in differences in lipid metabolism (Fig.3) should be discussed.

Thank you for noticing. We have added a few lines in the discussion section (lines 326-331) addressing the lack of differences in this lipid metabolism.

In the Discussion, no mentioned of the possible differences in shear stress derangement between TAV and BAV is given.

Thank you for this appreciation. We talk about hemodynamics alterations, as well as shear stress changes, in different parts of the discussion due to the abnormal aortic valve morphology (290-306, 361-368).

Also mitochondrial dysfunction and oxidative stress are also present in TAV as well as BAV.

Thank you for this comment. We have included a comment about this concept in the limitations section (page 13, lines 11-13).

Reviewer 2 Report

Comments and Suggestions for Authors

Bicuspid aortic valve (BAV) disease is associated with faster aortic valve degeneration and a high incidence of aortic stenosis (AS). In this study, authors aimed to identify differences in the pathophysiology of AS between BAV and tricuspid aortic valve (TAV) patients in a multi-omics study integrating metabolomics and transcriptomics, as well as clinical data.

Aortic valve tissue samples from AS patients with BAV disease have different metabolic and RNA profiles than those from TAV individuals. This suggests that the pathophysiology of AS differs between BAV individuals and TAV individuals. In BAV disease, AS was related to increased oxidative stress and mitochondrial dysfunction, which could be directly related to endothelial dysfunction and hemodynamic alterations in this disease. New therapeutic targets focusing on rescuing mitochondrial and endothelial function should be assessed to evaluate their utility in slowing aortic valve degeneration in BAV individuals.

The article is well written and deals with a fairly obvious topic, namely the different expression of messenger RNA in the development of bicuspid or tricuspid aortas. The fact that there are molecular targets has an important impact on medical culture, especially to be able to intervene on them.

Comments on the Quality of English Language

Bicuspid aortic valve (BAV) disease is associated with faster aortic valve degeneration and a high incidence of aortic stenosis (AS). In this study, authors aimed to identify differences in the pathophysiology of AS between BAV and tricuspid aortic valve (TAV) patients in a multi-omics study integrating metabolomics and transcriptomics, as well as clinical data.

Aortic valve tissue samples from AS patients with BAV disease have different metabolic and RNA profiles than those from TAV individuals. This suggests that the pathophysiology of AS differs between BAV individuals and TAV individuals. In BAV disease, AS was related to increased oxidative stress and mitochondrial dysfunction, which could be directly related to endothelial dysfunction and hemodynamic alterations in this disease. New therapeutic targets focusing on rescuing mitochondrial and endothelial function should be assessed to evaluate their utility in slowing aortic valve degeneration in BAV individuals.

The article is well written and deals with a fairly obvious topic, namely the different expression of messenger RNA in the development of bicuspid or tricuspid aortas. The fact that there are molecular targets has an important impact on medical culture, especially to be able to intervene on them.

Author Response

Thank you very much for your kind review. We think this paper could open the research of new therapeutic targets in BAV disease but also help understand better its development.